# Increase in the community circulation of ciprofloxacin-resistant Escherichia coli despite reduction in antibiotic prescriptions

Veronika Tchesnokova[1], Lydia Larson[1], Irina Basova[1], Yulia Sledneva[1], Debarati Choudhury[1], Thalia Solyanik[1], Jennifer Heng [1], Teresa Christina Bonilla[1], Sophia Pham[1], Ellen M. Schartz[2,3], Lawrence T. Madziwa[2,3], Erika Holden[2,3], Scott J. Weissman[4], James D. Ralston[2,3] & Evgeni V. Sokurenko [1✉]

## Abstract

**Background** Community circulating gut microbiota is the main reservoir for uropathogenic *Escherichia coli*, including those resistant to antibiotics. Ciprofloxacin had been the primary antibiotic prescribed for urinary tract infections, but its broad use has been discouraged and steadily declined since 2015. How this change in prescriptions affected the community circulation of ciprofloxacin-resistant *E. coli* is unknown.

**Methods** We determined the frequency of isolation and other characteristics of *E. coli* resistant to ciprofloxacin in 515 and 1604 *E. coli*-positive fecal samples collected in 2015 and 2021, respectively. The samples were obtained from non-antibiotic-taking women of age 50+ receiving care in the Kaiser Permanente Washington healthcare system.

**Results** Here we show that despite a nearly three-fold drop in the prescription of ciprofloxacin between 2015 and 2021, the rates of gut carriage of ciprofloxacin-resistant *E. coli* increased from 14.2 % to 19.8% (P = .004). This is driven by a significant increase of isolates from the pandemic multi-drug resistant clonal group ST1193 (1.7% to 4.2%; P = .009) and isolates with relatively few ciprofloxacin-resistance determining chromosomal mutations (2.3% to 7.4%; P = .00003). Though prevalence of isolates with the plasmid-associated ciprofloxacin resistance dropped (59.0% to 30.9%; P = 2.7E-06), the isolates co-resistance to third generation cephalosporins has increased from 14.1% to 31.5% (P = .002).

**Conclusions** Despite reduction in ciprofloxacin prescriptions, community circulation of the resistant uropathogenic *E. coli* increased with a rise of co-resistance to third generation cephalosporins. Thus, to reduce the rates of urinary tract infections refractory to antibiotic treatment, greater focus should be on controlling the resistant bacteria in gut microbiota.

## Plain language summary

The alarming rise of bacteria causing infections that are difficult to treat with antibiotics, known as multidrug-resistant bacteria, is a major problem in medicine. The reduction in the use of antibiotics has been encouraged to control the spread of antibiotic-resistant bacteria. Some multidrug-resistant bacteria reside in the gut of healthy individuals and can cause various forms of urinary tract infections (UTIs). Ciprofloxacin is an antibiotic that was widely used to treat UTIs, but strong recommendations to reduce its prescription have been recently introduced. We compared the presence of bacteria in the gut that could not be killed by ciprofloxacin in women aged 50 and above who do not use antibiotics and reside in the Seattle area. Despite a nearly three-fold drop in the prescription of ciprofloxacin between 2015 and 2021, antibiotic-resistant bacteria in the gut were found more frequently, affecting one in five women. Our study demonstrates that antibiotic-resistant bacteria continue to be present even when antibiotic prescriptions are reduced, demonstrating the need to undertake further similar studies.

[1] Department of Microbiology, University of Washington School of Medicine, 1705 NE Pacific St., Seattle, WA 98195, USA. [2] Kaiser Permanente Washington, 2715 Naches Ave. SW, Renton, WA 98057, USA. [3] Kaiser Permanente Washington Health Research Institute, 1730 Minor Ave, Suite 1600, Seattle, WA 98101-1466, USA. [4] Department of Laboratory Medicine, Seattle Children's Hospital, 1100 Olive Way Tutor Center, Seattle, WA 98101, USA. ✉email: evs@u.washington.edu

Antimicrobial resistance has reached pandemic proportions in the last few decades, increasing the mortality rates and healthcare costs associated with the prescription of ineffective antibiotics[1,2]. These developments prompted the establishment of antimicrobial stewardship programs in healthcare institutions to minimize the overuse of antibiotics and optimize appropriate antibiotic selection, dosing, and duration of therapy[3]. However, antibiotic-resistant bacteria are often circulating in the community as 'commensal' microbiota colonizing healthy individuals[4–6], and it remains unclear how antibiotic use reduction has affected their prevalence in that reservoir.

Urinary tract infections (UTIs)—bladder cystitis, pyelonephritis, and urosepsis of both community and nosocomial origin — are among the most common reasons for antibiotic treatment[7]. Women of postmenopausal age—generally 50 years or older (50+ yo)—are especially at high risk for severe and drug-resistant forms of UTI[5,8]. UTIs are primarily caused by extra-intestinal pathogenic *Escherichia coli* that are mostly associated with strains from specific clonal groups[9,10]. Resident gut bacteria carried by the patient as commensal microbiota are the primary reservoir of UTI-causing *E. coli*, including the drug-resistant strains[5,11–14].

Ciprofloxacin and other fluoroquinolones (FQ) had been, for many years, the most often prescribed antibiotics for UTI treatment and, commonly, for other infections[7,15]. Against *E. coli* infections resistant to FQ and other antibiotics, third-generation cephalosporins (3GC) are commonly used[16,17]. Wide use of FQ has led to the documentation of several concerning side effects, including neurotoxicity and tendinopathy, as well as the development of *Clostridiodes difficile* infections[18–20]. Most importantly, in the last two decades, there has been a rampant rise in FQ-resistant *E. coli* (FQREC) causing extra-intestinal infections and the FQREC occurrence has been strongly associated with the rates of hospitalization and mortality from sepsis[21]. The FQREC occurrence being strongly associated with the rates of hospitalization and mortality from sepsis[21]. The non-susceptibility to FQ is primarily associated with chromosomal point mutations in the quinolone resistance-determining regions (QRDR) coding the main targets of FQ—bacterial DNA topoisomerases GyrA (residues S83 and D87) and ParC (residues S80 and E84)[22]. A set of at least three QRDR mutations—at both positions 83 and 87 in GyrA and position 80 in ParC—typically results in the highest resistance level[23]. The FQR phenotype is also mediated by mobile, i.e., transferrable, elements, including chromosome-located transposons, insertion sequences, genomic islands, and, most often, plasmid-mediated quinolone resistance (PMQR) genes[24,25]. Genes carried on mobile genetic elements are also primarily responsible for resistance to 3GC, mostly mediated by extended-spectrum beta-lactamases (ESBL)[26].

To cutback the adverse effects associated with FQ and in an attempt to reduce the spread of resistance, a campaign was launched in the mid-2010s to avoid prescription in patients with uncomplicated UTI and reserve FQ for more severe forms of UTIs[19,27,28]. However, it remains questionable whether a reduction in antibiotic use can be effective in reducing the rates of resistance in *E. coli* infections[29]. Moreover, theoretical models suggest that, once emerged, the resistant isolates can keep circulating as commensal colonizers even in the absence of antibiotic use[30,31].

Here we examined the presence, clonal composition, FQR determinants, and 3GC co-resistance of gut colonizing FQREC isolated from fecal specimens collected during 2015 and 2021 from non-antibiotic taking women of age 50+ enrolled in the Kaiser Permanente Washington (KPWA) healthcare system. While the study was not designed for a direct association analysis, we describe a significant increase in gut colonization by FQREC

between 2015 and 2021 driven by a rise in specific bacterial lineages and 3GC co-resistance that occurred in parallel to the reduction in overall use of FQ but rise in the 3GC prescriptions among KPWA enrollees.

## Methods

**Study design and participants**. This study on the collection and analysis of fecal and urine samples from women without documented prescribed antibiotics was approved by the Kaiser Permanente Washington (KPWA) Research Institute Institutional Review Board and was originally carried out between July 2015 and January 2016, described here[5], and then repeated under the same protocol between May and December 2021. Study protocols included waivers of consent to identify potential participants. The risks and benefits of the research were explained to participants in a mailed invitation letter and consent information sheet. Participants provided informed consent for participation by sending a biological sample back in the mail. Sample collection took ~6 months during both studies because of the logistics of candidate selection, solicitation, mailing the kits, and receiving them back for analysis.

**Processing of fecal samples and typing of fecal *E. coli***. Sample processing and isolation, and typing of FQREC is described in detail in Supplemental Methods. Briefly, fecal samples were plated on either pre-poured HardyCHROM™ UTI agar plates (Hardy Diagnostic, USA) or plates prepared from HiChrome™ UTI Agar (HiMedia Laboratories Pvt, Ltd.) supplemented with ciprofloxacin at 0.5, 2.0, or 10.0 mg/L. Plates were incubated for 16–20 h at 37ºC, and up to 30 single colonies (SCs) morphologically identified as potential *E. coli* were cultured, saved, and tested further for (a) growth on ciprofloxacin-supplemented agar (0.5, 2.0, and 10.0 mg/L) and (b) clonality based on sequencing of four loci—*fumC, fimH, gyrA*, and *parC* (see Supplemental Methods for more detail). All single isolates that grew on plates with at least 0.5 mg/L of ciprofloxacin were termed FQREC and tested further for ciprofloxacin MIC and susceptibility to third-generation cephalosporins (3GC) (see Supplemental Methods for more detail). Additionally, subsets of FQREC were tested for the presence of PMQR, *bla*, and virulence factors determinants (see Supplemental Methods for more detail). The testing result are described in Supplemental Data 1. Primers are listed in Supplemental Data 7.

**Prescription data collection**. All KPWA enrollees (enrolled at least 10 months within a year) were selected for years 2010–2021, their EMRs were checked for prescription of FQ or 3GC class antibiotics at least once during the year, and the prescription rate was calculated as a percent of enrollees with at least 1 prescription per year from the total number of enrollees that year. The number of female enrollees who would have been 50+ yo in 2021 were identified, and their prescription rates were calculated separately.

**Statistical analysis**. The prevalence of FQREC and individual clones among 2015 and 2021 isolates was compared in the Chi-square test. Correlation between MIC level, number of QRDR mutations, and PMQR status was evaluated both in simple and in multiple linear regression models with Bonferroni corrections, as indicated. Trends in FQ and 3GC prescription rates were analyzed using linear regression, with *P* values reported. Changes in the level of resistance to antibiotics were evaluated for all isolates and for separate groups of isolates using logistic regression. Analysis was carried out using STATA 14.2 Software (StataCorp, Texas, USA). All throughout the manuscript, the mean statistics is accompanied by the standard deviation (±).

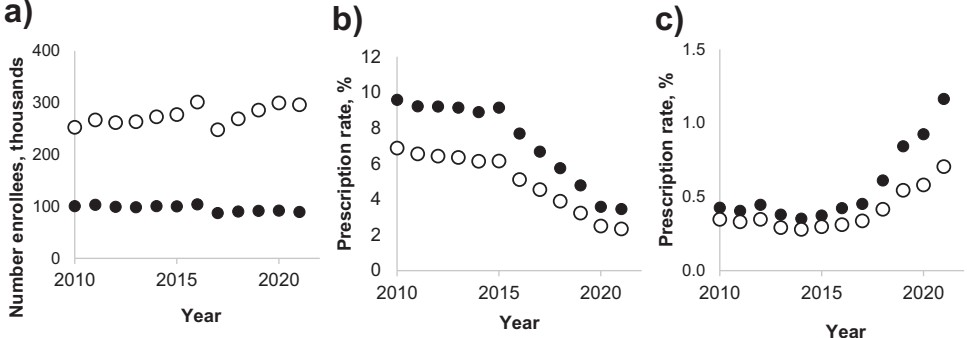

**Fig. 1 Antibiotics prescription among KPWA enrollees. a** Total enrollment of 18+ yo in KPWA per year. **b** FQ yearly prescription rates. **c** 3GC yearly prescription rates. Open circles—all enrollees, closed circles—female enrollees which would be 50+ years old in 2021. The prescription rates were calculated as the percent of people from the target population (all or only 50+ yo) enrolled that year who had at least one prescription of FQ (**b**) or 3GC (**c**) antibiotic within that year.

**Reporting summary**. Further information on research design is available in the Nature Portfolio Reporting Summary linked to this article.

## Results

**Decline in the prescription of FQ and concurrent rise in the prescription of 3GC**. Between 2010 and 2021, the number of enrollees in the Kaiser Permanente WA (KPWA) health plan (known as the GroupHealth Cooperative before 2018) of patients 18+ yo ranged between ~287 and ~339 K for total enrollment and ~98 to ~118 K for women aged 50+ yo (Fig. 1a). Between 2010 and 2015, the fraction of total enrollees who were prescribed FQ remained relatively steady (6.41 ± 0.28%), but from 2016 to 2021 the use of FQ declined >2.5-fold (to 2.35%, P = 0.000019) (Fig. 1b). Among the target study population—women who reached age 50+ by 2021—the use of FQ was significantly higher than in the total population in 2010–2015 (9.20 ± 0.39%, P = 3.25E-09) but also declined as drastically (to 3.40%, P = 0.000021) from 2016 to 2021. The use of 3GC was proportionally much less than that of FQ but prescription rates increased from 0.32 ± 0.03% in 2010–2015 to 0.71% by 2021 in the total population (P = 0.00029) and nearly threefold, from 0.40 ± 0.03% to 1.16%, in the target population (P = 0.00026) (Fig. 1c). The temporal changes in the FQ and 3GC use reflected more general trends observed among all Medicare Part D enrollees across USA and, more prominently, in the Washington state (Supplemental Fig. S1).

**The overall rate of gut carriage of FQREC increased due to a rise of certain isolate groups**. Among fecal samples submitted in 2015 and 2021 from women of age 50+, the vast majority (89.6 and 90.3%, respectively; P = 0.623) yielded *E. coli*. Among the *E. coli*-positive samples from the 2015 study, 14.2% contained FQREC (Table 1). In the 2021 study, the rate of FQREC-positive samples among the *E. coli* positive samples increased nearly 1.5-fold, to 19.8% (P = 0.004). In 46.9% (2015) and 55.0% (2021) fecal samples (P = 0.303) the number of colonies on the ciprofloxacin plate were indistinguishable by eye from that on the non-antibiotic plate, i.e., the FQREC bacteria were apparently predominant.

In 95.9 and 95.0% of FQREC-positive samples in 2015 and 2021, respectively, the resistant bacteria in each sample were clonally identical, i.e., belonged to the same clonal group. The two most common FQR clonal groups remained the same in both studies—the pandemic multidrug-resistant ST131-*H30* and ST1193 (Table 1). While the combined carriage of the two pandemic clonal groups rose only <u>insignificantly</u>, from 7.6% in

**Table 1 Presence, clonality, and QRDR mutations profile of fecal FQR *E. coli* isolated in 2015 and 2021.**

| Category | 2015 | | 2021 | | P value[b] |
|---|---|---|---|---|---|
| | No | %[a] | No | %[a] | |
| **Total** | 575 | | 1778 | | |
| *E. coli*[a] | 515 | 89.6 | 1604 | 90.2 | 0.652 |
| FQR *E. coli*[c] | 73 | 14.2 | 318 | 19.8 | **0.004[e]** |
| ST131-*H30* | 30 | 5.8 | 70[f] | 4.4 | 0.174 |
| ST1193 | 9 | 1.7 | 68 | 4.2 | **0.009** |
| ST69[d] | | | | | |
| Total | 7 | 1.4 | 51 | 3.2 | **0.028** |
| ≥3 QRDR | 4 | 0.8 | 8 | 0.5 | 0.466 |
| 2 QRDR | 3 | 0.6 | 29 | 1.8 | **0.047** |
| 1 QRDR | 0 | 0.0 | 4 | 0.2 | 0.257 |
| 0 QRDR | 0 | 0.0 | 10 | 0.6 | 0.073 |
| Other[d] | | | | | |
| Total | 28 | 5.4 | 140 | 8.7 | **0.016** |
| ≥3 QRDR | 20 | 3.9 | 66 | 4.1 | 0.817 |
| 2 QRDR | 1 | 0.2 | 5 | 0.3 | 0.822 |
| 1 QRDR | 6 | 1.2 | 47 | 2.9 | **0.026** |
| 0 QRDR | 2 | 0.4 | 24 | 1.5 | **0.047** |

[a]For "*E. coli*" category, percent from all samples is given; for all other categories percent from only all *E. coli*-containing samples is calculated.
[b]Difference in prevalence between samples from the 2015 and 2021 study was evaluated using the Chi square test for 2 × 2 tables; P values <0.05 are in bold.
[c]Three samples in the 2015 collection (one sample with two clones—H30 and other ST, and two samples with four and two other ST clones) and 16 samples in the 2021 collection (one sample contained both ST131-H30 and ST1193, 2 samples contained ST131-H30 and ST69, three samples contained ST1193 and ST69, one sample contained ST1193 and other FQR clone, three samples contained ST69 and other FQR clone and six samples contained a mix of other FQR clones).
[d]Among other STs both 2015 and 2021 collections had ST131-H41, ST38, ST405, ST457, ST648, ST1163, as well as several STs from Clonal Complexes CC10, CC21, CC58; additionally, 2015 collection had isolates from ST156, ST410, and CC1196; 2021 collection had isolates belonging to ST95, CC14, ST636, ST349, ST394, ST354, etc. ST69 group represents an ST69 clonal complex.
[e]The increase in the prevalence of FQREC-positive samples was statistically significant for all samples as well (12.7 vs 17.9% in 2015 and 2021, respectively, P = 0.003).
[f]One ST131-H30 has only 1 QRDR mutation at GyrA-83.

2015 to 8.6% in 2021 (P = 0.462), carriage of ST1193 increased ~2.5-fold (P = 0.009) and became almost equal to that of the previously dominant ST131-*H30*. All 2015 and 2021 isolates of ST1193 and, with one exception, ST131-*H30* had a full set of at least three QRDR mutations—two in GyrA and one in ParC.

The third most common FQREC in both studies were strains from one of the largest uropathogenic clonal groups, ST69, but its carriage rate increased >2-fold between 2015 and 2021 (P = 0.028) (Table 1). In contrast to ST131-*H30* and ST1193, only 15.7% of the 2021 ST69 isolates had ≥3 QRDR mutations and the ST69 overall increase was almost exclusively due to

**Table 2 Distribution of plasmid-mediated quinolone resistance (PMQR) determinants among FQR *E. coli* isolates with different numbers of QRDR mutations.**

| No. QRDR mutations | Presence of PMQR determinants | 2015 | | 2021 | | P value[a] |
|---|---|---|---|---|---|---|
| | | No. isolates | % | No. isolates | % | |
| no QRDR | YES | 2 | 100 | 34 | 94.4 | 0.732 |
| | NO | 0 | | 2 | | |
| 1 QRDR | YES | 3 | 50 | 13 | 25.5 | 0.206 |
| | NO | 3 | | 38 | | |
| 2 QRDR | YES | 1 | 25 | 5 | 13.9 | 0.555 |
| | NO | 3 | | 31 | | |
| ≥3 QRDR | YES | 40 | 60.6 | 47 | 22.4 | **5.53E-09** |
| | NO | 26 | | 163 | | |
| Total | YES | 46 | 59 | 99 | 29.7 | **1.14E-06** |
| | NO | 32 | | 234 | | |

[a]Presence of PMQR in 2015 vs 2021 collections was compared separately for *E. coli* with different numbers of QRDR mutations using Chi-square test, with *P* values < .05 indicated in bold.

isolates with only two QRDR mutations (S83L of GyrA and either S80I or E84 of ParC) ($P = 0.047$).

The remainder of FQREC belonged to various smaller (mostly uropathogenic) clonal groups (a total of 22 in 2015 and 65 in 2021, Supplemental Table S1), with the combined carriage rate of such isolates increasing significantly in 2021 ($P = 0.016$). Notably, the increase of the FQREC from the smaller clonal groups was driven by a significant rise of isolates with either a single mutation in GyrA ($P = 0.026$) or no QRDR mutation at all ($P = 0.047$) (Table 1).

Based on a random set of FQREC from different clonal groups, the majority of them contained one or more genes coding potential urovirulence factors (see Supplemental Table S2).

**Isolates with FQ resistance-mediated mobile elements became less prevalent.** Irrespectively of the QRDR mutations number, almost two-thirds of 2015 FQREC carried the PMQR determinants (Table 2). In contrast, in 2021, isolates overall carriage of the PMQR determinants dropped ~2-fold ($P = 1.14E-06$). Though the PMQR determinants remained (nearly) omnipresent among 2021 isolates with no QRDR mutations and the drop among the isolates with either one or two QRDR mutations was insignificant, the decline among the isolates with ≥3 QRDR mutations was pronounced (from 60.6 to 22.4%; $P = 5.53E-09$). In both years, most of the isolates with the PMQR determinants contained *qnrB* alone, followed by *qnrS* and, then, *aac(6')-Ib-cr* (Supplemental Table S3).

**Increase in the prevalence of FQREC with a lower resistance level to FQ.** We investigated whether the gut FQREC isolates from 2015 and 2021 differ in the minimal inhibitory concentration to FQ by using the broth dilution method (range 0.5–8.0 mg/L, see Methods) (Table 3). The overall proportion of FQREC isolates unable to grow at >1 mg/L increased from 5% in 2015 to 20% in 2021, while those able to grow >8 mg/L decreased from 87 to 67% ($P = 0.0008$). We investigated next how the presence of chromosomal QRDR mutations and PMQR determinants affected the ability of FQREC to grow at different ciprofloxacin concentrations. The determined MIC level directly correlated overall with the number of QRDR mutations ($P = 2.39E-111$): >99% of strains with ≥3 mutations but <5% of isolates with <3 mutations were able to grow at >8 mg/L. Moreover, the majority of isolates with no or one mutation (67%) were unable to grow at >1 mg/L. Since nearly all isolates with no QRDR mutations had the PMQR determinants and nearly all strains with the full set of mutations had MIC above 8 mg/L, we used only a subset of isolates (with 1–2 QRDR mutations) to evaluate an association between the

presence of PMQR determinants and the MIC level (Table 3). Indeed, there was a strong correlation between higher MIC and the presence of PMQR determinants based on the linear regression analysis ($P = 0.004$). After controlling for the number of QRDR mutations and the presence of PMQR determinants in a multiple linear regression model, the difference in MIC observed originally between the 2015 and 2021 isolates lost its statistical significance ($P = 0.411$).

**Resistance to 3GC has increased among the gut FQREC.** The overall gut carriage rate of *E. coli* co-resistant to FQ and 3GC rose >3-fold between 2015 and 2021 (2.1 vs 6.6%; $P = 0.0001$, with their proportion among FQREC isolates rising from 14.1 to 31.5% ($P = 0.002$) (Fig. 2). The 3GC resistance increased across different categories of FQREC, reaching 40% among ST131-H30 isolates and above 30% among isolates from non-pandemic clonal groups and those with ≥3 QRDR mutations. Only among ST1193 isolates did the 3GC resistance rate still remain below 20%. Among the 3GC co-resistant isolates from either 2015 or 2021, the majority were ESBL producers (90.9 vs. 80.2%, respectively; $P = 0.386$). Among all identified beta-lactamases, 65% of isolates harbored *bla*$_{CTX-M}$ determinants (Supplemental Table S4). The total resistance profiles of FQREC to other antibiotics are shown in Supplemental Table S5, with the 2021 isolates exhibiting an expected major increase in the resistance to one-generation cephalosporins but a decrease in the resistance to trimethoprim/sulfamethoxazole.

**Discussion**

The commensal gut microbiota is the primary reservoir of UTI-causing *E. coli*, with gut carriage profiles mirroring UTI from the perspectives of clonal composition and drug-resistance profiles of *E. coli* as well as patient age demographics[32,33]. In particular, elderly women are colonized more often by FQREC and are at a higher risk for FQR forms of UTI than younger women[5]. As recently as 2016, FQs were the most prescribed antibiotic for treating UTIs[34]. However, accumulated evidence for adverse effects combined with a sharp rise in FQ resistance of uropathogenic enterobacteria led to FDA, CDC, and IDSA recommendations for curbing FQ use in the treatment of uncomplicated UTI[19,27,28,35,36]. However, FQ continues to be recommended for patients who have no alternative treatment options, complicated UTIs, pyelonephritis, or other severe infections where the benefits of FQ outweigh the risks of adverse effects[17,20,36]. Thus, resistance to FQ in uropathogens remains a significant clinical problem, and a reduction in gut carriage of FQREC would be a welcome development. This is especially

**Table 3 Relationship between ciprofloxacin MIC and study year, number of QRDR mutations, and presence of PMQR determinants in FQR *E. coli*.**

| No. QRDR mutations | Group | Total no. isolates | MIC (mg/L) | | | | | P value |
|---|---|---|---|---|---|---|---|---|
| | | | ≤1 | 2 | 4 | 8 | >8 | |
| Total | 2015 | 77 | 4 (5) | 3 (4) | 3 (4) | 0 (0) | 67 (87) | 0.000007[a] |
| | 2021 | 315 | 62 (20) | 29 (9) | 10 (3) | 4 (1) | 210 (67) | |
| ≥3 QRDR | | 274 | 0 (0) | 0 (0) | 1 (0) | 1 (0) | 272 (99) | 2.39E-111[b] |
| 2 QRDR | | 32 | 8 (25) | 15 (47) | 6 (19) | 1 (3) | 2 (6) | |
| 1 QRDR | | 52 | 32 (62) | 14 (27) | 3 (6) | 1 (2) | 2 (4) | |
| no QRDR | | 34 | 26 (76) | 3 (9) | 3 (9) | 1 (3) | 1 (3) | |
| 1-2 QRDR | PMQR+ | 22 | 7 (32) | 5 (23) | 5 (23) | 1 (5) | 4 (18) | 0.004[a] |
| | PMQR- | 62 | 33 (53) | 24 (39) | 4 (6) | 1 (2) | 0 (0) | |

[a]Here, a decrease in MIC between 2015 and 2021 and PMQR-positive and PMQR-negative FQREC was evaluated using simple linear regression.
[b]Increased MIC association with a higher number of QRDR mutations was assessed in a multiple linear regression model using the number of mutations as a continuous variable and adjusting for the presence of PMQR determinants

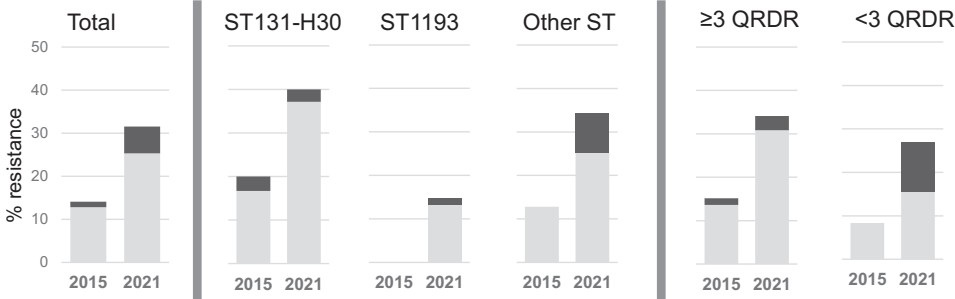

**Fig. 2 Proportion of 3GC-resistant strains among FQREC.** Distribution of total 3GC-resistance (dark gray) and ESBL-carriers (light) among FQR *E. coli* isolates from 2015 and 2021 *E. coli* collections, total, belonging to different clones (ST131-*H30*, ST1193, and all others), or harboring different number of QRDR mutations.

important because the FQR phenotype of *E. coli* has a strong association with UTI-caused bacteremia in individuals aged 50+ yo and overall sepsis mortality rates in adults[37,38].

Here we show that the use of FQ was significantly higher among women of age 50+ compared to all enrollees in the KPWA healthcare system, potentially reflecting the higher incidence of UTIs in this category of women. In both populations, however, FQ prescription significantly declined between 2015 and 2021. While our study was not designed to establish a direct causal correlation between the changes in antibiotics use and resistance level, certain temporal changes in the profiles of gut FQREC might be potentially linked to the drop in FQ use. These include the higher prevalence of gut *E. coli* with a reduced (but still clinically relevant) resistance level (due to the presence of fewer QRDR mutations) and the overall reduction in isolates carrying the resistance-mediating genes carried on mobile genetic elements. Surprisingly, however, the total rate of gut carriage of fully resistant clonal groups with a complete set of the QRDR mutations remained at least the same, and the prevalence of FQREC from the pandemic multidrug-resistant clonal group ST1193 significantly increased. As a result, the rate of the overall gut carriage of FQR *E. coli* significantly increased in 50+ yo women between 2015 and 2021, despite the decline in FQ prescriptions. This and the fact that women in our study did not use any antibiotics for at least one year before providing the fecal sample suggests that community circulation of gut colonizing FQREC can be fully sustained even in the absence of antibiotic use, consistent with some predictions[30,31].

For the last 20 years, clonal group ST131-*H30* has been globally predominant among FQREC and multidrug-resistant *E. coli*[8,13,39]. Starting a decade ago, however, another UTI-associated clonal group of FQREC - ST1193 - has been on the rise world-wide[40,41]. We show here that the ST1193 gut carriage rate went up between 2015 and 2021 and is now on par with ST131-*H30*. It is worrisome because ST1193 appears to be more pathogenic than ST131-*H30* as it more frequently causes UTIs in younger women with a relatively robust host defense system[11,42]. It is unclear whether the higher virulence or other fitness attributes are linked to the significant rise in the gut carriage, but if ST1193 continues to increase in the coming years, the result could be an increase in infections caused by it even if the tendency to reduce prescriptions of FQ continues. Similarly, another FQREC clonal group that is on the rise in the gut carriage, ST69, is known to more frequently cause UTI in children to whom it could be transmitted[43,44]. The extensive genetic diversity of UTI isolates does not allow us to determine with confidence whether the fecal isolates and clinical urinary isolates from the same clonal group have similar uropathogenic potential. However, fecal FQREC in our study tends to have at least one of the typical urovirulence factors and the high-resolution clonal identity of isolates that is achieved by the *fumC-fimH* (CH) typing has previously been shown to be a good predictor of the uropathogenic nature of the isolates[10,45–48].

Bacterial topoisomerases GyrA and ParC are the main targets of FQ, and the acquisition of QRDR mutations is the major mechanism of *E. coli* becoming non-susceptible to FQ[49]. It is provocative to suggest that reduced usage of FQ could create a selective environment allowing FQREC strains with just a few QRDR mutations to circulate in greater numbers. However, the rise in the prevalence of FQREC from ST69 was due to isolates that carry two out of three QRDR mutations— one in GyrA and one in ParC (see Supplemental Table S6). This combination of

mutations has been rarely observed in previous studies, and it was proposed that, upon the acquisition of the first mutation in GyrA, there is a weak resistance gain by a mutation in ParC without the simultaneous occurrence of a second mutation in GyrA[22,50–52]. Importantly, a simultaneous double mutation is a very low-probability genetic event, which should be a limiting factor for the emergence of the fully resistant FQREC during the treatment with high FQ doses[23]. Thus, the rise of FQREC with two QRDR mutations could present a significant clinical problem, because the two-mutation strains are only one mutation away from becoming highly resistant. This simple genetic event could be easily selected in a patient during antibiotic therapy, resulting in treatment failure and infection relapse. Further studies are needed, however, to determine the clinical significance of the two-mutation strains and the genetic backgrounds underlying their rise among the ST69 strains.

The prevalence of isolates carrying mobile PMQR genes was reduced between 2015 and 2021, suggesting that the selective pressure needed to maintain the plasmids became weaker. However, this reduction was noted only among isolates with QRDR mutations, while carriage of the mobile resistance genes remained essential for the FQREC isolates without the mutations. While the impact of the mobile resistance elements is less pronounced quantitatively than that of the QRDR mutations, considering the rise of FQREC without QRDR mutations between 2015 and 2021, the role of mobile elements in the FQ non-susceptibility remains clinically significant.

Finally, there was an increase in gut carriage of FQREC strains that are also resistant to 3GC. The primary but not exclusive mechanism of 3GC resistance in *E. coli* is the production of ESBL, which is a diverse class of enzymes dominated by the CTX-M family typically coded by plasmid-associated genes[53]. In our analysis, 3GC resistance increased across all FQREC categories and clonal groups, suggesting that this had happened under a generally imposed selective pressure. Especially troublesome is the doubling of 3GC resistance among the isolates with ≥3 QRDR mutations, i.e., highly resistant to FQ. Notably, the prescription of 3GC was significantly increased in KPWA enrollees between 2015–2021, especially among women 50+ yo. The reason for the increase is not fully understood but may potentially be associated with 3GC being used more often as a replacement choice for FQ. Though a direct correlation between the rise in the 3GC co-resistance and increased use of the antibiotic was not investigated here, it is plausible to suggest that these are interconnected developments. The only drop in antibiotic resistance was, to some extent, noted for trimethoprim/sulfamethoxazole. It remains unclear, however, whether this could be explained by a possible change in the use of this antibiotic (not analyzed here), co-carriage of the resistance genes with PMQR determinants, or anything else.

Taken together, the results of our study suggest that, while increased use of antibiotics in patients can lead to the emergence of resistant isolates, the latter can continue to spread in the community even if antibiotic use is decreased. In turn, the continuous level of community circulation of the resistant bacteria could lead to a sustained rate of antimicrobial therapy-refractory infections, consistent with a recent study[29]. Therefore, on the one hand, our study fully supports the efforts of antimicrobial stewardship to decrease the overuse of antibiotics. On the other hand, a reduction in specific antibiotic prescriptions may not alone be a sufficient measure to reduce the spread of resistance. Another factor to consider is the use of FQ in animals like poultry farming and its spread into the environment, that was not accounted for in our study but might have a significant effect on the circulation of resistant bacteria in the community[54]. One way or another, our study suggests that besides the reduction of antibiotic use,

identification and/or selective decolonization of carriers may provide an effective way to control resistance in clinical infections. The decolonization measures could be especially effective if implemented in high-risk patients or community groups and/or against the most clinically dangerous resistant strains. This might be achieved, for example, via the use of probiotic strains, bacteriophages, or both[55–58]. However, a better understanding of both the epidemiology and ecology of antibiotic-resistant bacteria is needed to identify the basis of their fitness as colonizers and the best way to target them. Thus, we propose that there may be a need to expand the role of antimicrobial stewardship programs from being focused only on antibiotic use in clinical settings to being also oriented towards screening for and decolonization of commensal carriage of antibiotic-resistant strains in the most vulnerable individuals.

## Data availability

The authors confirm that the data supporting the findings of this study are available within the article and its supplementary materials. Supplemental Data 1 contains the list of FQREC from 2015 and 2021 fecal samples with the results of any testing performed on them (sequencing, antimicrobial resistance, PMQR, BLA, and VF determinants, number and type of QRDR mutations in GyrA and ParC and ciprofloxacin MIC. These data were used in Tables 1–3, S1–S6, and Fig. 2. Sanger sequences of *fumC* and *fimH* genes of FQREC were submitted to CHtyper[59] (https://cge.food.dtu.dk/services/CHTyper/) for allele identification; typing results are in the Supplemental data 1 file. Sanger sequences of *gyrA* and *parC* genes of FQREC were aligned and unique sequences were assigned allele numbers using the in-house allele databases, with *.fasta files supplied as Supplemental Data 2 and Supplemental Data 3 files; typing results are listed in Supplemental Data 1 file. Supplemental Data 4 file contains the numerical data used to make Fig. 1: total number of enrollees in KPWA and number of enrollees prescribed FQ and 3GC antibiotics, for 2010–2021 by year. Supplemental Data 5 file contains the numerical data used to make Supplemental Fig. S1: total number of Medicare enrollees in the USA and in Washington state and number of beneficiaries with claims for prescribed FQ and 3GC antibiotics, for 2013–2020 by year. Supplemental Data 6 file contains data used in Table 1: a list of all samples from 2015 and 2021 studies with information regarding the presence of any *E. coli* or FQREC in them. Supplemental Data 7 file contains the list of all primers used in this study, including those published elsewhere, with appropriate references in the Supplemental Information file.

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

## Acknowledgements

We would like to thank Drs. Steve Moseley, Paul Thottingal, Jessica Zering, Katarina Kameran, Marisa D'Angeli, and Yuan-Po Tu for the critical reading of the manuscript and suggestions for additional data to improve it; Bianca DiJulio and KPWHRI Survey Research Program for outstanding organization and implementation of the survey; Arina Pushkina, Denys Davydenko, and Isaac Pasumansky for excellent technical assistance. This work was supported by the National Institutes of Health R01AI106007 and R01AI150152 to E.V.S.

## Author contributions

E.V.S., J.D.R., and V.T. designed the research; E.M.S. designed and implemented the survey for KPWARHI, with the help of L.T.M., E.H. managed the project for KPWA and KPWARHI, V.T. managed the project for UW; L.L., Y.S., I.B., D.C., T.S., T.C.B, J.H., and S.P. processed samples and helped with data analysis; V.T. performed final data analysis and statistical analysis, with the help of E.V.S. and S.J.W. V.T. and E.V.S. wrote the manuscript, with the help of J.D.R., S.J.W., and all other authors. All authors approved the manuscript.

## Competing interests

The authors declare no competing interests.
