## [Peer Review File · Communications Medicine]

Reviewers' comments:

Reviewer #1 (Remarks to the Author):

General:

This is an interesting paper demonstrating the probable increase in FQREC carried asymptotically by adults in the Kaiser system in the Pacific Northwest. The strengths are the disturbing increasing trend in these putative pathogens, especially ST1193. The weaknesses are in data/methodology incompleteness, and the limited conclusions regarding trends based on 2 ~ 6 month sampling intervals occurring 6 years apart.

Minor comments:

1. Line 87: A better term is "tendinopathy"
2. Line 88: "Clostridioides"
3. Results: What does +/- mean? SD?
4. Line 136. Presumably the rate of FQREC-positive isolates also significantly differed among all participants' stools (not just those colonized with *E. coli*). Please confirm.
5. Table 1. I might have missed it, but I think that the number of positives relates not to the FQREC, but to the number of women harboring such organisms.

Major comments:

1. The title and the text infer that there is an increase in gut carriage of the putatively pathogenic FQREC of all clonal types, and that ST1193 is supplanting ST131-H30. That might be true, and that data suggest those changes, but the authors only two ~ 6 month intervals (lines 306-7) were studied. That I
2. Line 310: what is the composition and commercial origin of the "UTI agar" plates?
3. Prior studies of gut-colonizing fluoroquinolone-resistant *E. coli* in children, including a cohort from Seattle, reported the susceptibilities of the isolates to non-fluoroquinolone antibiotics beyond cefoxitin and/or the complement of extra-intestinal virulence loci (PMID: 25969564, PMID: 17005812). These data would be useful in this paper as well, perhaps on a random subset of the 394 such isolates in Table 1.
4. The authors infer that the FQREC are uropathogens, but some corroboration of their virulence potential beyond two-locus sequence typing would be useful. Again, a random subset could be analyzed if of sufficient numbers.
5. Line 316: what antibiotics does "HardyCHROM ESBL HDx 316 (Hardy Diagnostics, USA)" contain?
6. Line 311: the authors state "Plates were incubated 16- 20h at 37oC, and up to 30 single colonies (SCs) morphologically identified as potential *E. coli* were cultured, saved, tested for ESBL production, and typed" Are these isolates pre-selected by plating on agar containing ciprofloxacin?
7. What percent of the presumptive *E. coli* on the UTI agar plate were quinolone resistant? In prior studies, 2 or 3 or more of 5 colonies were resistant.
8. Line 134: how were the *E. coli* in general (not pathogen specific) defined? Presumably by a phenotype on an agar plate. Please provide details, and confirm that lactose fermentation is not part of the definition as ST1193 are generally lactose nonfermenting.

Reviewer #2 (Remarks to the Author):

Overall the papers findings are novel, interesting, and clinically impactful, as this research highlights the issue of co-resistance in the selection of AMR mechanisms within clinically relevant bacteria. Historically it has been reported that for certain resistance mechanisms, removal of the selection leads to a decrease (over time) of that type of resistance in pathogens in our community. However, there now seems to be less of a fitness cost for these mechanisms in dominant pathogenic lineages, and in the case of 3GC-R organisms specifically, a strong link between the carriage of multiple resistance determinants/mutations within dominating MDR associated pathogenic lineages.

I do have some main points about the methods and the description of your results - to assess 'resistance level', you grew isolates on agar containing varying concentrations of ciprofloxacin. However it is not described in your methods which protocol you used to do this. Was this agar susceptibility test in line with CLSI standards? Susceptibility testing must be carried out on standardized MHAI agar - this is especially important when assessing the susceptibility of fluoroquinolones such as ciprofloxacin as they are zwitterionic in nature, therefore they can chelate cations. If you do not use a cation adjusted agar such as MHAI, your susceptibility results would be skewed. Moreover, you try to determine the relationship between the presence of quinolone resistance-determining regions and the level of antibiotic non-susceptibility. This analysis is also not sufficient to measure associations between the number of resistance mutations and antibiotic non-susceptibility - agar susceptibility testing is not considered 'gold standard', and to understand the link between resistance mutations and antimicrobial non-susceptibility, broth microdilution in line with CLSI standards must be carried out on these isolates. A table having the MIC range, and the # of isolates recorded with each MIC value with various numbers of QRDR would have been more impactful.

An additional table describing the types of QRDR mutations identified and their prevalence across the most common MLST types would have also been useful - it is not just the number of mutations present that can impact fluoroquinolone susceptibility, but also the type/combination of types of mutations present can impact fluoroquinolone susceptibility.

For your statistical analysis (fisher's test) - did you apply a Bonferroni correction? You carried out multiple pairwise tests on a large number of isolates, therefore it is important to adjust for sample number to prevent aberrant results.

Reviewer #3 (Remarks to the Author):

This paper by Tchesnokova and colleagues describes the increase in gut carriage of important fluoroquinolone resistant E. coli clones from 2015 to 2021 despite the reduced use of fluoroquinolone usage clinically. Surveillance of the carriage of antibiotic resistant bacteria is important to understand difficult to treat pathogens that may be shed in the hospital environment and that are circulating in the community. Further, these studies are important for informing antibiotic prescribing. The statistical analyses are valid and the work reproducible. However, the clarity of the manuscript could be improved and below are minor comments and suggestions for modifications that will, in this reviewer's opinion, strengthen this paper.

- Have consistency throughout the manuscript when describing age groups: ie. ≥ 18 yo OR 18+ yo

- Line 58: Clarify here and throughout manuscript: “genes carried on mobile genetic elements” versus “mobile genes”
- Line 57 to 60: Consider revising into 2 sentences. The first part talks about resistance genes and the second part talks about a phenotype.
- Line 90 to 91: This sentence is incomplete/awkward. Consider merging with the previous sentence.
- Line 93: Why “so-called”? Here is the chance to list the specific mutations that are discussed later.
- Line 121: Consider changing “but since” to “from 2016 to 2021”
- Line 125: Consider changing “by 2021” to “from 2016 to 2021”
- Line 132. Consider clarifying the title to: “Increased rate of FQREC gut carriage in women ≥ 50 yo due to rise of specific clonal groups”.
- Line 138 is unclear. Please re-write and clarifying this sentence. In the 2015 dataset the FQR ST131 accounts for 42% of FQR, not 95.9%?
- Describe the 3 QRDR mutations in the intro (which SNPs?).
- Line 157: Consider using the \leq symbol here and throughout manuscript for clarity
- Line 158: If there is only 1 mutation or 0 mutations does that account for FQR resistance? The phenotypic results are later on in the manuscript but consider explaining these isolates are still non-susceptible to cipro.
- Line 191: Which ESBL genes did the authors identify. A sentence here about the specific ESBL genes found would be informative.
- Line 257 to 259: Are PMQR genes always on plasmids or can then be on IS elements?
- Line 306: The 2015 isolates in this study are the same from the ones analysed in ref #5?
- Line 501: In the title please clarify that this is from the ≥ 50 yo target population
- There are 2 sets of Figure legends in the manuscript. The y-axis on Figure 1a needs to be corrected (not 300 enrollees but 300,000). 1b and 1c – these are % rates per total prescriptions of the total population and target population?
- Figure 2 legend: Please clarify. Should this say “ESBL carriers”?

May 25, 2023

To: Dr. Katharine Barnes
Editor
Communications Medicine

Dear Dr. Barnes,

Below we describe the changes that were made to our manuscript, Increase in the Rate of Gut Carriage of Fluoroquinolone-Resistant Escherichia coli despite a Reduction in Antibiotic Prescriptions, in response to the reviewers' comments and concerns.

Respectfully,

Veronika Tchesnokova

Evgeni V. Sokurenko

Response to the reviewers' comments

Reviewer #1 (Remarks to the Author):

General:

This is an interesting paper demonstrating the probable increase in FQREC carried asymptotically by adults in the Kaiser system in the Pacific Northwest. The strengths are the disturbing increasing trend in these putative pathogens, especially ST1193. The weaknesses are in data/methodology incompleteness, and the limited conclusions regarding trends based on 2 ~ 6 month sampling intervals occurring 6 years apart.

Minor comments:

1. Line 87: A better term is "tendinopathy"

Answer. Thank you for suggestion, we have made the change.

2. Line 88: "Clostridioides"

Answer. Thank you, we have changed to correct name.

3. Results: What does +/- mean? SD?

Answer. Yes, that is correct. We have added this notion in the end of the Statistical methods section.

4. Line 136. Presumably the rate of FQREC-positive isolates also significantly differed among all participants' stools (not just those colonized with *E. coli*). Please confirm.

Answer. Yes, that is correct, the P value was 0.004. We have added this information as a footnote to Table 1.

5. Table 1. I might have missed it, but I think that the number of positives relates not to the FQREC, but to the number of women harboring such organisms.

Answer. Yes, that is correct. We have rephrased the sentence to clarify.

Major comments:

1. The title and the text infer that there is an increase in gut carriage of the putatively pathogenic FQREC of all clonal types, and that ST1193 is supplanting ST131-H30. That might be true, and that data suggest those changes, but the authors only two ~ 6-month intervals (lines 306-7) were studied.

Answer. Thank you for bringing up this point. We have now clarified in the Methods section that the sample collection period lasted approximately six months for both studies. This duration was necessary due to the logistics involved in participant selection, kit solicitation, mailing the kits, and receiving them back for analysis.

Additionally, in the discussion section, we now emphasize that our study is cross-sectional rather than longitudinal in nature. We want to clarify that we do not have information about the clonal dynamics between the two samplings in particular individuals or within the six-month period of each sampling. Despite this limitation, we firmly believe that the fundamental methodology and conclusions of the analysis remain valid for the comparison of the 2015 and 2021 studies.

2. Line 310: what is the composition and commercial origin of the "UTI agar" plates?

Answer. We now include in Methods and Supplemental Methods that, for agar plating of fecal samples without antibiotics, we used pre-poured HardyCHROM™ UTI agar plates (Hardy Diagnostic, USA), and for isolation of FQ or 3gCS resistant *E. coli* we prepared plates containing the appropriate antibiotics using HiChrome™ UTI Agar (HiMedia Laboratories Pvt, Ltd.). Both UTI agars have a proprietary composition aimed at differentiation uropathogenic microorganisms based on the colonies' color, thus their specific components remain unknown to us.

3. Prior studies of gut-colonizing fluoroquinolone-resistant *E. coli* in children, including a cohort from Seattle, reported the susceptibilities of the isolates to non-fluoroquinolone antibiotics beyond cefoxitin and/or the complement of extra-intestinal virulence loci (PMID: 25969564, PMID: 17005812). These data would be useful in this paper as well, perhaps on a random subset of the 394 such isolates in Table 1.

Answer. As part of the revision, we conducted measurements of antibiotic non-susceptibility using a standard antibiotics panel. We have included the results as a supplemental table (Supplemental Table S4) which provides the cumulative antibiogram.

In addition, we assessed the presence or absence of six virulence factors (*gadA/B*, *sat*, *iha*, *senB*, *vat*, and *ireA*) for a subset of isolates. These factors were selected based on a recent previous study (PMC6325179). We have presented the corresponding data in Supplemental Table S2.

4. The authors infer that the FQREC are uropathogens, but some corroboration of their virulence potential beyond two-locus sequence typing would be useful. Again, a random subset could be analyzed if of sufficient numbers.

Answer. As mentioned above, we have sequenced six putative urovirulence factors in a subset of isolates. We also indicated in the discussion now that, while the extensive genetic diversity of UTI isolates does not allow to pinpoint with confidence which factors are the best predictors of urovirulence potential, the high-resolution clonal identity of isolates that is achieved by the *fumC-fimH* (CH) typing has previously been shown to be a good predictor of the uropathogenic nature of the isolates (PMID: 22226951, 23843485, 24041881, 28077674, 27001705, 32645942, etc.).

5. Line 316: what antibiotics does “HardyCHROM ESBL HDx 316 (Hardy Diagnostics, USA)” contain?

Answer: Ceftazidime and cefpodoxime. We have added this information in the Supplemental Methods section.

6. Line 311: the authors state “Plates were incubated 16- 20h at 37oC, and up to 30 single colonies (SCs) morphologically identified as potential *E. coli* were cultured, saved, tested for ESBL production, and typed” Are these isolates pre-selected by plating on agar containing ciprofloxacin?

Answer: We now describe in more detail how the samples were processed and isolates were picked in Supplemental Methods section so that it would be clear that indeed isolates were pre-selected on agar with ciprofloxacin. In brief, each sample was plated on several UTI agar plates – plain and containing ciprofloxacin at different concentrations – and from each plate colonies appearing to be *E. coli* (magenta or pink or, rarely, opaque color), were picked, saved and analyzed.

7. What percent of the presumptive *E. coli* on the UTI agar plate were quinolone resistant? In prior studies, 2 or 3 or more of 5 colonies were resistant.

Answer: Due to our study design (plating in parallel on plain UTI and ciprofloxacin-UTI agar) we can only do a rough (by eye) estimation of the colonies number of total *E. coli* vs FQREC. We can estimate that in 46.9% (2015) and 55.0% (2021) fecal samples (P = .303) the number of colonies on ciprofloxacin plate were indistinguishable by eye from

that on the non-antibiotic plate, i.e. the FQREC bacteria were apparently predominant. We have added this information to the Results section.

8. Line 134: how were the *E. coli* in general (not pathogen specific) defined? Presumably by a phenotype on an agar plate. Please provide details, and confirm that lactose fermentation is not part of the definition as ST1193 are generally lactose nonfermenting.

Answer: Reviewer is correct in that we have initially picked potential *E. coli* based on their color on the UTI agar plate as per manufacturer instructions, including those belonging to ST1193 that develop magenta color colonies. The exact chemical composition and reactivity of the chromogenic compounds is not provided by the manufacturer and we now clarify that lactose fermentation per se does not appear being the only criterion.

Reviewer #2 (Remarks to the Author):

Overall the papers findings are novel, interesting, and clinically impactful, as this research highlights the issue of co-resistance in the selection of AMR mechanisms within clinically relevant bacteria. Historically it has been reported that for certain resistance mechanisms, removal of the selection leads to a decrease (over time) of that type of resistance in pathogens in our community. However, there now seems to be less of a fitness cost for these mechanisms in dominant pathogenic lineages, and in the case of 3GC-R organisms specifically, a strong link between the carriage of multiple resistance determinants/mutations within dominating MDR associated pathogenic lineages.

Comment 1: I do have some main points about the methods and the description of your results - to assess 'resistance level', you grew isolates on agar containing varying concentrations of ciprofloxacin. However, it is not described in your methods which protocol you used to do this. Was this agar susceptibility test in line with CLSI standards? Susceptibility testing must be carried out on standardized MHAll agar - this is especially important when assessing the susceptibility of fluoroquinolones such as ciprofloxacin as they are zwitterionic in nature, therefore they can chelate cations. If you do not use a cation adjusted agar such as MHAll, your susceptibility results would be skewed. Moreover, you try to determine the relationship between the presence of quinolone resistance-determining regions and the level of antibiotic non-susceptibility. This analysis is also not sufficient to measure associations between the number of resistance mutations and antibiotic non-susceptibility - agar susceptibility testing is not considered 'gold standard', and to understand the link between resistance mutations and antimicrobial non-susceptibility, broth microdilution in line with CLSI standards must be carried out on these isolates. A table having the MIC range, and the # of isolates

recorded with each MIC value with various numbers of QRDR would have been more impactful.

Answer: As part of the revision, we have determined the ciprofloxacin MIC using broth microdilution method as per CLSI procedure. The previous conclusion about correlation between the number of QRDR mutations and MIC remained valid. We indicated now that isolates harboring 1-2 QRDR mutations we were the only set where we could assess the effect of PMQR and that we found a direct association between higher MIC values and presence of PMQR determinants. We have replaced Table 3 in the manuscript using the MIC data instead of plating on CIP-agar plates **and re-written the appropriate section of the Results.**

Comment 2: An additional table describing the types of QRDR mutations identified and their prevalence across the most common MLST types would have also been useful - it is not just the number of mutations present that can impact fluoroquinolone susceptibility, but also the type/combination of types of mutations present can impact fluoroquinolone susceptibility.

Answer: We have added now Supplemental Table S6 which lists the GyrA-ParC combinations found in our study and the number of isolates within major FQREC groups that have each combination.

Comment 3: For your statistical analysis (fisher's test) - did you apply a Bonferroni correction? You carried out multiple pairwise tests on a large number of isolates, therefore it is important to adjust for sample number to prevent aberrant results.

Answer: Yes, we indicated now that we have used the Bonferroni correction every time multiple comparisons are done on the same dataset. We have expanded the Statistical analysis section of the Methods and also added footnotes in the Tables wherever appropriate.

Reviewer #3 (Remarks to the Author):

This paper by Tchesnokova and colleagues describes the increase in gut carriage of important fluoroquinolone resistant E. coli clones from 2015 to 2021 despite the reduced use of fluoroquinolone usage clinically. Surveillance of the carriage of antibiotic resistant bacteria is important to understand difficult to treat pathogens that may be shed in the hospital environment and that are circulating in the community. Further, these studies are important for informing antibiotic prescribing. The statistical analyses are valid and the work reproducible. However, the clarity of the manuscript could be improved and below are minor comments and suggestions for modifications that will, in this reviewer's opinion, strengthen this paper.

- Have consistency throughout the manuscript when describing age groups: ie. ≥ 18 yo OR 18+ yo

Answer: Thank you, we have corrected everywhere to 18+ yo and 50+ yo.

- Line 58: Clarify here and throughout manuscript: “genes carried on mobile genetic elements” versus “mobile genes”

Answer: Thank you, we have corrected this throughout the manuscript

- Line 57 to 60: Consider revising into 2 sentences. The first part talks about resistance genes and the second part talks about a phenotype.

Answer. We modified the sentence as follows: *Prevalence of the fluoroquinolone resistance-associated genes carried on mobile genetic elements among the resistant isolates dropped from 59.0% to 30.9% ($P<.001$). At the same time, their co-resistance to third generation cephalosporins has increased 14.1% to 31.5%, $P=.002$).*

- Line 90 to 91: This sentence is incomplete/awkward. Consider merging with the previous sentence.

Answer: We modified the sentences as follows: *Most importantly, in the last two decades there has been a rampant rise in FQ resistant *E. coli* (FQREC) causing extra-intestinal infections and FQREC occurrence has been strongly associated with the rates of hospitalization and mortality from sepsis²¹.*

- Line 93: Why “so-called”? Here is the chance to list the specific mutations that are discussed later.

Answer: As suggested, we have removed ‘so-called’ and listed the positions for QRDR mutations in GyrA and ParC.

- Line 121: Consider changing “but since” to “from 2016 to 2021”

Answer: We have corrected this per suggestion of the reviewer.

- Line 125: Consider changing “by 2021” to “from 2016 to 2021”

Answer: We have corrected this per suggestion of the reviewer.

- Line 132. Consider clarifying the title to: “Increased rate of FQREC gut carriage in women ≥ 50 yo due to rise of specific clonal groups”.

Answer: We agree that the suggested title would be technically more specific, but we strongly believe that the current title reflects better the main significance of our study that there is an overall rise in gut carriage of FQREC that occurred despite a reduction in antibiotic use. While expanding the title by adding the suggested specific information would be of value, unfortunately we are already at the limit (slightly above actually) of the title length.

- Line 138 is unclear. Please re-write and clarifying this sentence. In the 2015 dataset the FQR ST131 accounts for 42% of FQR, not 95.9%?

Answer. We modified the sentence as follows: *In 95.9% and 95.0% of FQREC-positive samples in 2015 and 2021, respectively, the resistant bacteria in each sample were clonally identical, i.e. belonged to the same clonal group.*

- Describe the 3 QRDR mutations in the intro (which SNPs?).

Answer: Per reviewer's suggestion we have now listed the QRDR SNPs in GyrA and in ParC in the Introduction (lines 96-97).

- Line 157: Consider using the \leq symbol here and throughout manuscript for clarity

Answer: We have replaced 'fewer than three' with <3 throughout the manuscript.

- Line 158: If there is only 1 mutation or 0 mutations does that account for FQR resistance? The phenotypic results are later on in the manuscript but consider explaining these isolates are still non-susceptible to cipro.

Answer: We revised the sentence as follows: *Notably, the increase of the FQREC from the smaller clonal groups was driven by a significant rise of isolates with either a single mutation in GyrA ($P=.026$) or no such mutation at all ($P=.047$) (Table 1).*

- Line 191: Which ESBL genes did the authors identify. A sentence here about the specific ESBL genes found would be informative.

Answer: As part of the revision, we have carried out analysis of the presence of major beta-lactamase loci in the 3GCR isolates and have included the data as Supplemental Table S4. We used 5 multiplex reactions described in the paper by Dallenne et al (PMID: 20071363). The major determinant of non-susceptibility to third generation cephalosporins in FQREC were CTX-M beta-lactamases (65% of isolates with identified beta-lactamases carried them). Additionally, 3GCR isolated carried TEM, OXA-1, LAT, ACC-1, FOX and CIT beta-lactamase determinants. Overall, in 84% of tested 3GCR isolates at least one beta-lactamase locus determinant was found, with 39.3% of those carrying more than one determinant; in 16% cases no genes potentially responsible for the ESBL phenotype could be identified.

- Line 257 to 259: Are PMQR genes always on plasmids or can then be on IS elements?

Answer: We agree that the latter could be the case and added a sentence now that the PMQR genes include those that may be potentially located on any mobile elements, including chromosome-located transposons, insertion sequences, genomic islands, etc.

- Line 306: The 2015 isolates in this study are the same from the ones analysed in ref #5?

Answer: Yes, that is correct, and we clarified that the study in reference #5 analyzed all women 18+ yo, but in the current study we have only included a subset from that study comprised of women 50+ yo.

- Line 501: In the title please clarify that this is from the ≥ 50 yo target population

Answer: As we mentioned above, while we agree with the reviewer about a value of being more specific in the title, adding there that our target population were specifically ≥ 50 yo women would further exceed the allowed word count and require removing wording about the general points that we find critical. However, we specified the target population in the upper portion of the abstract and indicated in the beginning of Introduction that it has been very important for us to focus specifically on women of age 50+ because of the high risk of drug-resistant UTI in this population.

- There are 2 sets of Figure legends in the manuscript. The y-axis on Figure 1a needs to be corrected (not 300 enrollees but 300,000). 1b and 1c – these are % rates per total prescriptions of the total population and target population?

Answer: Thank you for the comment. We have removed Figure Legends from the Figure files, leaving it only in the main manuscript file at the end. We have corrected the Y-axis in Figure 1 and described that the rates are calculated as percent of people from target population (all or only 50+ yo) enrolled that year who had at least one prescription of FQ (B) or 3GC (C) antibiotic within that year.

- Figure 2 legend: Please clarify. Should this say “ESBL carriers”?

Answer: Yes, thank you, we have corrected this.

REVIEWERS' COMMENTS:

Reviewer #2 (Remarks to the Author):

The edits made satisfy the recommendations made by myself and the other reviewers. This paper provides a current insight into the changing mechanistic basis of fluoroquinolone resistance among a predominant uropathogen. The continued study of the epidemiology of uropathogenic bacteria, as well as the associated trends in resistance and characterization of resistance determinants is paramount to support the clinical management of urinary tract infections.